



# Spatial parameter optimization of a terrestrial biosphere model for improving estimation of carbon fluxes for deciduous forests in the eastern United States: an efficient model-data fusion method

Rui Ma[1], Jingfeng Xiao[2], Shunlin Liang[3], Han Ma[1], Tao He[1], Da Guo[4], Xiaobang Liu[1], Haibo Lu[5]

[1] School of Remote Sensing and Information Engineering, Wuhan University, Wuhan 430079, China

[2] Earth Systems Research Center, Institute for the Study of Earth, Oceans, and Space, University of New Hampshire, Durham, NH 03824, USA

[3] Department of Geographical Sciences, University of Maryland, College Park, MD 20742, USA

[4] College of Resources and Environment, University of Chinese Academy of Sciences, Beijing 100049, China

[5] Guangdong Province Key Laboratory for Climate Change and Natural Disaster Studies, School of Atmospheric Sciences, Sun Yat-sen University, Guangzhou, China

*Correspondence to*: slliang@whu.edu.cn

**Abstract.** Inaccurate parameter estimation is a significant source of uncertainty in complex terrestrial biosphere models. Model parameters may have large spatial variability, even within a vegetation type. Model uncertainty from parameter uncertainty

can be significantly reduced by model-data fusion (MDF), which, however, is difficult to implement over a large region with traditional methods due to the high computational cost. This study proposed a hybrid modeling approach that couples a terrestrial biosphere model with a data-driven machine learning method, which uses satellite information and considers the physical mechanisms. We developed a two-step framework to estimate the essential parameters of the revised Integrated Biosphere Simulator (IBIS) pixel by pixel using the satellite-derived leaf area index (LAI) and gross primary productivity

(GPP) products as "true values." The first step was to estimate the optimal parameters for each sample using a modified adaptive surrogate modeling algorithm (MASM). We applied the Gaussian Process Regression algorithm (GPR) as a surrogate model to learn the relationship between model parameters and errors. In our second step, we built an eXtreme Gradient Boosting (XGBoost) model between the optimized parameters and local environmental variables. The trained XGBoost model was then used to predict optimal parameters spatially across the deciduous forests in the eastern United States. The results

showed that the parameters were highly variable spatially and quite different from the default values over forests, and the simulation errors of the GPP and LAI could be markedly reduced with the optimized parameters. The effectiveness of the optimized model in estimating GPP, ecosystem respiration (ER) and net ecosystem exchange (NEE) were also tested through site validation. The optimized model reduced the root-mean-squared-error (RMSE) from 7.03 to 6.22 gC m$^{-2}$ 8d$^{-1}$ for GPP, 2.65 to 2.11 gC m$^{-2}$ 8d$^{-1}$ for ER and 4.45 to 4.38 gC m$^{-2}$ 8d$^{-1}$ for NEE. The mean annual GPP, ER and NEE of the region from

2000 to 2019 were 5.79, 4.60 and -1.19 Pg year$^{-1}$, respectively. The strategy used in this study requires only a few hundred model runs to calibrate regional parameters and is readily applicable to other complex terrestrial biosphere models with





different spatial resolutions. Our study also emphasizes the necessity of pixel-level parameter calibration and the value of remote sensing products for per-pixel parameter optimization.

# 1 Introduction

Accurate quantification of the terrestrial carbon budget is crucial for understanding the global carbon cycle and biosphere-atmosphere interactions and informing climate projections (Fernández-Martínez et al., 2018; Piao et al., 2020). Land-surface models (LSMs) built on process-based mechanisms of atmosphere-biosphere interactions are often used to simulate the behavior of the terrestrial carbon cycle in response to a changing climate (Peaucelle et al., 2019). These models typically use a large number of parameters. Prescribed values of model parameters based on a theoretical assumption, empirical analysis, or

field measurements are prone to substantial uncertainties which can engender inaccurate projections in modeling (Barman et al., 2014; Keenan et al., 2012; Kuppel et al., 2014). Parameter estimation is becoming more challenging, especially when greater details are introduced to enhance the authenticity and interpretability of LSMs (Famiglietti et al., 2021).

Model-data fusion (MDF) is an increasingly used method that can be leveraged to reduce the model-data misfit by calibrating parameters. Researchers have mainly used the site and sample measurements, raw reflectivity observations, and satellite-based

products to estimate parameters in complex terrestrial biosphere models, and significant progress has been made in the integration of univariate observations (MacBean et al., 2016). However, due to the complex relationship between model variables, univariate assimilation is often not enough to constrain the vegetation parameters involved in multiple surface processes, and it is still necessary to introduce other data streams to increase the constraints on model parameters (Fernández-Martínez et al., 2018; Liu et al., 2015; Schürmann et al., 2016; Zobitz et al., 2014). For example, soil moisture observations

can be regarded as an additional constraint of a plant functional type (PFT) due to the tight coupling of carbon and water cycles in vegetation photosynthesis (Scholze et al., 2016), and the fraction of photosynthetically active radiation data (FAPAR) and leaf area index (LAI) could bring phenology information in constraining long-term vegetation dynamics. Joint constraints of LAI/FAPAR and in-situ observations of carbon fluxes or atmospheric $CO_2$ concentration have already been explored in reducing parameters uncertainties (Forkel et al., 2014; Bacour et al., 2015).

The parameter estimation of complex models has been well studied at the site scale. Many researchers used observations to optimize parameters at the site scale and then apply the optimized parameters to regions. However, when a PFT covers a broad area, ecosystem characteristics, density, or disturbance history can vary substantially across the region. In this case, site-scale parameters with a small footprint cannot be considered spatially representative, and the relationship between observations and models may not be spatially scalable (Raoult et al., 2016; Xiao et al., 2014; Zhou et al., 2020). Finding a globally applicable

parameter scheme is complicated for LSMs, as it requires a high computational cost (Gong et al., 2017). Machine learning (ML) approaches are flexible in adapting to an increasing stream of geospatial products, making it easy to extract patterns and combine them with physical models as an additional source of information. It offers an opportunity to improve or accelerate parameterizations by integrating model simulation and multiple observations or high-quality spatial products more intensively





in different ways. For example, Chaney et al. (2016) carried out parameter calibration at 85 eddy covariance flux sites and then obtained the spatial parameters using extra-trees and combining local environmental characteristics. Although the parameters showed significant spatial variability, their study did not verify the performance of the optimized model. Tao et al. (2020) constrained parameters at the site scale to improve the soil organic carbon simulation and then extended the optimized parameters to the United States utilizing a neural network. Their results showed that the model error was significantly reduced when the spatial heterogeneity of optimal parameters was considered. These studies using flux sites to calibrate parameters can ensure the parameter accuracy at each site, while it is likely to cause potential problems of overfitting due to limited training samples, which makes it difficult to guarantee the accuracy when extending the resulting optimal parameters to a broad region using ML.

To date, few researchers conducted pixel-level parameterization because parameter optimization often depends on a large number of parameter samplings and model operations, especially when using the Markov Chain Monte Carlo (MCMC) method as the optimization algorithm. MCMC is usually applied to model parameter calibration to obtain the optimal posterior probability distribution of parameters (Yuan et al., 2012; Safta et al., 2015). It typically requires thousands of model simulations, which are excessively expensive for complex LSMs that may take hours for each model simulation (Fer et al., 2018). ML can be an innovative method to conduct surrogate modeling or emulation in parameter optimization (Li et al., 2018a; Reichstein et al., 2019). ML is regarded as an effective method to speed up model parameterizations, and can make better use of the abundant spatial information provided by the multisource high-precision remote sensing products. Researchers have proved the feasibility of surrogate modeling and indicated that once the ML emulator is trained well, the optimization speed can be increased by an order of magnitude without much loss of accuracy (Koike et al., 2014; Fer et al., 2018). For example, Gong et al. (2017) pointed out that using a surrogate model can reduce the number of model runs from 10^6 to a few hundred or thousands with acceptable accuracy. The integration of physical and ML models may not only achieve improved performance but may also significantly improve computing speed, and the optimization of model parameters using multisource observations based on surrogate modeling has the potential for improving the accuracy of regional or global LSMs (Qiuru Zhang 2019; Xu et al., 2018).

Some researchers considered finding unknown regional parameters using an ML emulator but with a very coarse resolution (Dagon et al., 2020; Li et al., 2018a). This is because a large number of model runs are still needed to train the surrogate model to ensure its accuracy, even though adaptive surrogate modeling-based optimization algorithms have been developed to reduce the initial training samples (Li et al., 2014; Gong et al., 2017). In this paper, we explored a two-step method of combining ML and a physical model to improve the calibration speed of spatial parameters, made full use of high-quality remote sensing products to calibrate the model at each pixel, and carried out a study on the deciduous forests (DF) in the eastern United States. We first performed pixel-level parameter calibration using surrogate running instead of a terrestrial model within samples in our region. We then expanded the optimal parameters obtained from samples into spatial distribution using the ML approach with several environmental variables.




This paper is organized as follows. Section 2 briefly introduces a terrestrial model, and related data sources. Section 3 describes the two-step pixel-by-pixel region parametrization algorithm and the experimental setting. Section 4 shows the results of thus optimizing parameters and presents a spatial analysis of carbon fluxes before and after the optimization.

Uncertainty analysis and future work are described in Section 5, followed by a conclusion in Section 6.

## 2 Models and Data

### 2.1 Model framework and running strategy

#### 2.1.1 Model description

The Integrated Biosphere Simulator (IBIS) model integrates many land-surface ecosystem processes into a complex physical

mechanism model, which is divided into the land-surface, vegetation phenology, carbon balance, and vegetation dynamic modules (Foley et al., 1996; Yuan et al., 2014). It has a hierarchical structure operating on 60-min or one-year time steps. IBIS considers three snow layers and six soil layers in each pixel and determines PFTs based on different ecological characteristics. Detailed information about IBIS is available in Foley et al. (1996) and Kucharik et al. (2000).

A simple function is used in IBIS to describe the relationship between leaf behaviors and phenological status, which forecasts

the onset of budburst when suitable temperatures are reached. It describes the defoliation when the 10-day average daily air temperature is below 5°C in autumn. The potential LAI ($pLAI$), also known as the yearly maximum leaf area, is calculated for the growing season based on the leaf carbon biomass of previous years, as shown in Eq. (1). When simplified phenology ($phen$, shown below and calculated by the accumulated growing degree day and several temperature parameters) is considered for $pLAI$, the overall growth dynamics of vegetation leaves are obtained for each year (Cao et al., 2015):

$$pLAI = leaf\ biomass \times specla, \tag{1}$$

$$LAI = pLAI \times phen, \tag{2}$$

where $specla$ refers to specific leaf area ($m^2\ kg^{-1}$).

Biomass allocation in the IBIS model is updated annually, and therefore it is difficult to accurately describe the influence of meteorological factors on LAI and to capture the detailed dynamics of leaves (Kucharik et al., 2006). Capturing the daily dynamics of LAI requires changes in the leaf biomass pool each day. For this process, we incorporated part of the carbon allocation in the Data Assimilation Linked Ecosystem Carbon (DALEC) model (Chuter et al., 2015) into the original IBIS

model, a simple box model evaluating six-carbon pools for deciduous forests daily. The photosynthesis rate of deciduous forests conveyed following the Farquhar equation (Farquhar et al., 1980) can be expressed as the minimum of light and Rubisco limited rates of photosynthesis in the IBIS model, and then we can obtain gross primary productivity (GPP) at each time step on a pixel basis. Photosynthates stored in daily GPP will be allocated to the foliage ($C_f$), woody ($C_w$), and root ($C_r$) pools within a discrete dynamical process as defined in the DALEC model. The dynamic LAI values can be obtained by dividing daily $C_f$

by the leaf mass per area. LAI values most directly limit specific growth or carbon allocation characteristics, potentially





influencing the seasonal dynamics of carbon-related fluxes. Net ecosystem exchange (NEE) is defined as the difference between ecosystem respiration (ER) and GPP. In this paper, a positive value of NEE represents carbon release, while a negative value indicates carbon uptake.

### 2.1.2 Running strategy

We simulated the LAI and carbon fluxes for the deciduous forests in the eastern United States from 2000 to 2019 at a spatial resolution of $0.05° \times 0.05°$. All model simulations were forced with daily climate drivers, $CO_2$ concentration, and soil properties. We employed fixed vegetation instead of dynamic runs to avoid model errors from building forest species structures while simulating ecological processes. All the datasets used to drive the model are described in Section 2.2.

For model initialization, the improved IBIS was cyclically spun up for 50 years by repeating meteorological datasets in 2000.
The final soil and carbon states of initialization run at each pixel reached a quasi-equilibrium and were saved as inputs for the subsequent transient simulation. Default values for the initial carbon pools (six pools for DF) used in DALEC modeling were set to the same scheme as Chuter et al. (2015).

### 2.2 Data sources

### 2.2.1 Model forcing

Initial soil and vegetation properties and daily climate data are required in IBIS. The soil parameters include soil sand and clay content (%), which were gathered from available comprehensive, gridded Global Soil Datasets for use in Earth System Models (GSDE) with a resolution of 30 arc-seconds (Shangguan et al., 2014). Forcing meteorological datasets are maximum and minimum temperature (°C), precipitation (mm), wind speed (ms$^{-1}$), specific humidity (%), pressure (Pa), and cloud cover (%). All forcing datasets were resampled to match the model resolutions (0.05°) using bilinear resampling, and hourly pressure was
aggregated into daily values. The global $CO_2$ concentration was based on the measurements at the Mauna Loa Observatory, Hawaii started in 2000 (Thoning et al., 1989). Information on all the datasets is summarized in Table 1.

The primary vegetation type in each pixel was identified by the high-resolution land cover maps from the National Land Cover Database (NLCD, 30 m) (Homer et al., 2020, available on www.mrlc.gov). The NLCD offers a suite of the national land cover of the United States and associates changes that were upscaled to the model running resolution based on the majority
land cover type (> 50%) method in resampling. We extracted data on deciduous broad-leaved forests in the eastern United States with the resampled land cover map.

### 2.2.2 Assimilated remotely sensed products

This study used global LAI and GPP products from the Global Land Surface Satellite (GLASS) suite as "observations" ("true values") for parameter calibration on a spatial scale. This dataset has long-time coverage, high spatial resolution (500m and
0.05°), and temporal and spatial continuity. It was found to have high accuracy in a site validation and product comparison





test (Liang et al., 2021) and has been widely used in many studies (Chen et al., 2019; Ryu et al., 2018; Kumar et al., 2020). We used the latest version of the GLASS LAI product, which was obtained from the bidirectional Long Short-Term Memory deep learning model (Ma et al., 2021). According to the validation by ground measurements, the overall accuracy of this product was much improved ($R^2$ = 0.73, RMSE = 0.82 $m^2$ $m^{-2}$) over the Moderate Resolution Imaging Spectroradiometer

(MODIS) LAI product ($R^2$ = 0.57, RMSE = 1.08 $m^2$ $m^{-2}$) (Ma et al., 2021). The GLASS GPP product with 0.05° spatial resolution was derived from a revised light-use efficiency model (EC-LUE model), which had a better performance at most sites ($R^2$ = 0.81, RMSE = 2.13 gC $m^{-2}$ $d^{-1}$) than FLUXCOM and several process-based biophysical models (Zheng et al., 2020).

### 2.2.3 Flux observations

We collected 14 eddy covariance flux sites of deciduous forests in the eastern United States from FLUXNET2015 ("Tier 1")

(Pastorello et al., 2020, https://fluxnet.org/data/fluxnet2015-dataset/) and AmeriFlux network (http://public.ornl.gov/ameriflux) (Table S1). Half-hourly or hourly measurements from AmeriFlux Datasets were aggregated into different temporal scales. Daily values were indicated as invalid if more than 20% of the data were missing on a given day. Site data from FLUXNET2015 were filtered considering quality information flags provided by the FULLSET data product (NEE_VUT_REF_QC ≥ 0.5) (Yuan et al., 2012).

### 3 Model-data fusion for pixel-level parameterization

### 3.1 Modified adaptive surrogate modeling (MASM)

We employed adaptive surrogate modeling in this study to establish a fast and practical framework for the iterative optimization of a complex physical model. We slightly improved the Adaptive Surrogate Modeling-based Optimization Parameter Optimization and Distribution Estimation (ASMO-PODE) method proposed by Gong et al. (2017). We applied Bayesian

optimization twice to reduce the time cost of Monte Carlo iterations. The modified procedure (Fig. S1) is as follows:

*Step 1: Initial random sampling*

We generated points in a uniformly distributed parameter space by randomly sampling using the Good Lattice Points approach with ranked Gram-Schmidt decorrelation (Owen, 1994). As Wang et al. (2014) pointed out, the ASMO method performs better when the number of initial points is set to 15–20 times the number of the parameters; we chose 18 as our initial

sampling multiple. With the sampled points, the dynamic model (IBIS) was run to obtain the corresponding target variables (daily LAI and GPP, as $y_t^M$ in Eq. (3)).

$$y_t^M = f_M(x_t, \theta, ini) + R \tag{3}$$

where $y_t^M$ is the modeled target variables (e.g., LAI, GPP) at time t, $f_M$ is the IBIS model, $x_t$ represents the forcing datasets (Table 1), $\theta$ represents the parameters, *ini* represents the initial soil and carbon pools after spinning up, and *R* represents the model errors.





*Step 2: Cost function (CF) and surrogate model construction*

We used the GLASS products (see Section 2.2.2) as the "observations" (i.e., "true values") to implement pixel-level parameter schemes to calibrate the model parameters. Under the observation constraints, the posterior probability distribution of parameters can be obtained with Bayes' rules and is proportional to the likelihood and prior probability density:

$$P(\theta_i|y_t^O) \propto P(y_t^O|\theta_i)P(\theta) = L(\theta_i|y_t^O)P(\theta_i) \quad \theta \sim uniform(range); \; i = 1,2,3,\cdots,n \qquad (4)$$

where $y_t^O$ is the observation, $P(\theta_i|y)$ denotes the posterior probability density functions of selected parameters given by the $y$, $P(y|\theta_i)$ means likelihood, and $P(\theta_i)$ ($i = 1,2,3,\cdots,n$) represents prior distribution (set as uniform), and $n$ is the number of sensitive parameters. The distribution bounds of each parameter are shown in Table 2. We aggregated the daily variables simulated by the model into 8 days and constructed the CF with observations.

$$L(\theta_i|y_t^O) = (2\pi\sigma)^{-n/2} \prod_t exp \left\{ -\frac{[y_t^O-y_t^M]^2}{2\sigma^2} \right\}, \qquad (5)$$

where $\sigma^2$ is the error variance of each observation. We assumed that the errors between the model and the observation ($y_t^O - y_t^M$) are independent (i.e., the covariance is zero), and therefore $\sigma^2$ here is expressed by the variance of each observation product (Yuan et al., 2012). We used *-2log(L)* as our final target function.

$$CF_t^S = f_s (CF_t^M, \theta_{sam}), \qquad (7)$$

where $CF_t^S$ stands for CF estimated from the GPR model ($f_s$), $CF_t^M$ stands for CF estimated from IBIS model ($f_M$), and $\theta_{sam}$ denotes sampled parameters.

*Step 3: Surrogate optimization protocol based on the MASM algorithm*

We performed global optimization on the GPR model using MASM, which requires many iterative operations that cost much less than running the highly complex IBIS model. One slight difference between MASM and the ASMO-PODE is that we applied two Bayesian optimizations for the GPR model. The first one (Fig. S1, Bayes1) was mainly the same as Gong's method (Gong et al., 2017), but we reduced the number of Monte Carlo iterations and set it to 10,000 to establish a relatively high-precision GPR model. The second application (Fig. S1, Bayes2) was aimed mainly at the repeated iteration of the GPR model obtained in the previous step. After running Bayes1, we calculated the error between the CF estimated by the GPR model ($CF_t^S$) and the original model ($CF_t^M$), as shown by Eq. (8).

At the end of each Monte Carlo iteration in Bayes1, five representative samples were selected adaptively from the posterior distribution of parameters and were then used as inputs to the IBIS model to obtain the corresponding $CF_t^M$. A new sample scheme containing those representative points was used to update the GPR model again. This procedure significantly reduced the running time of the initial model and obtained a relatively high-precision surrogate model when the first stop criterion was reached (Niter = 200). In the second optimization (Bayes2), we repeated the iterative optimization of the finally established GPR model and added the Gelman and Rubin (GR) factor shown in Eq. (9) as the index to judge the convergence of the Markov chain. When the GR factor was less than 1.2, the Markov chain was considered to have converged and the result was considered credible and repeatable (Gelman and Rubin, 1992). The model was run for both initial sampling (Fig. S1, Run1)





and adaptive resampling (Fig. S1, Run2), and the total number of model evaluations was Npars × 18 + 5 × Niter. The model

was run in parallel using MATLAB. Compared with the parameter calibration for the original model, this number was reduced

by an order of magnitude, which greatly reduced the computational resources.

$$RMSE = \sqrt{\frac{1}{n}\sum_{n=0}^{n}(CF_t^S - CF_t^M)^2} \tag{8}$$

$$GR = \sqrt{\frac{N_2-1}{N_2} + \frac{Nchain+1}{Nchain \times N_2} \times \frac{B}{W}}, \tag{9}$$

where *B* represents the variance between the averaged values of each chain, and *W* is the average of the variances within each

chain.

**3.2 Experimental design for generating pixel-level parameter estimates**

**3.2.1 Parameter screening**

Adding a new module inevitably introduces more parameters (Section 2.1.1), resulting in higher computational costs. Before

calibration, we needed to screen parameters and select the ones with the most influence on the target variables for calibration.

A few parameters related to photosynthesis, carbon allocation, respiration, and water stress were considered in our revised

IBIS model. Table 2 lists all parameters, together with the prior ranges and their descriptions. Some parameters are independent

of vegetation types by default (e.g., alpha3 and theta3), while those related to turnover time or allocation are generally regarded

to vary by type of vegetation (e.g., tauleaf, aleaf, and aroot). We randomly chose one-tenth of the deciduous forest pixels

(~4200 pixels) and then adopted the Morris approach (Morris, 1991) to roughly remove parameters with low influence on our

target variables (Fig. 1). We finally determined 10 sensitive parameters, consisting of alpha3, theta3, beta3, vmax, p5, p14,

p15, p17, p18, and p20, for further calibration.

**3.2.2 Model-data fusion on samples**

We randomly selected another sample (~4200 pixels) in the deciduous forest ranges of the eastern United States. The model

was run for 20 years between 2000 and 2019, with 2000 used as the spin-up year and 2001–2009 as the calibration period. The

observed and simulated values of the remaining time (years after 2009) were used for independent verification.

A pixel-by-pixel optimization was performed stepwise for each sample using the MASM method with the GPP and LAI

from the GLASS products as the observed values. The daily GPP simulated by the model was aggregated into 8 days, and the

MASM method was used for iterative optimization to obtain the posterior probability distribution of four sensitive parameters

(alpha3, theta3, beta3, and vmax), which showed a higher sensitivity of GPP. The mean values of those parameters were taken

as the fixed values for LAI optimization in the second step. Adopting such a stepwise optimization required the use of a

"parameter block" approach so that each data stream could optimize only the more strongly correlated parameters (Wutzler





and Carvalhais, 2014). We tested the effect of optimization order on a small number of samples, and the results showed no significant differences. A multivariable stepwise optimization performed by Alton (2013) also showed that optimization order did not distinctly change the results.

### 3.2.3 Spatial expansion using the XGBoost approach

The optimal parameter scheme of each sample was obtained after calibration. Considering the spatial differences in the
environmental characteristics, the machine learning (ML) method was used to expand the optimal scheme of the samples. We used the eXtreme Gradient Boosting (XGBoost) algorithm to describe the nonlinear relationship between parameters and environment variables, which has achieved great success in ML and has been widely used in remote sensing classification, surface variable inversion, and information extraction (Zhong et al., 2019; Pilaš et al., 2020; Liu et al., 2021).

Similar to sample running, the regional simulation was also forced with multisource driving datasets (Table 1) and adopted
the same spin-up scheme to reach equilibrium. We ran the spatial scale parameters predicted by XGBoost back into the physical model and verified the final accuracy. After obtaining a parameter distribution with a resolution of 0.05° on the spatial scale, we estimated the LAI, GPP, ER, and NEE for the deciduous forests of the eastern United States.

### 3.3 Error evaluation

The evaluation in this paper involves three aspects: (1) verification of the accuracy of MASM when calibrating parameters;
(2) verification of the feasibility of the XGBoost method in terms of predicting parameters; (3) verification of the accuracy in spatial scale and multi-product comparisons. We used Pearson's correlation coefficient (R), root-mean-squared-error (RMSE), average error (AE), and a comprehensive index called the Distance between Indices of Simulation and Observation (DISO) (Zhou et al., 2021; Hu et al., 2019) as statistical measures. AE can specify whether errors are overestimated or underestimated. RMSE is more sensitive to large errors. DISO used here is a combination of three widely used statistical metrics (R, normalized
RMSE, and AE), aiming to identify the overall evaluation of each verification. When the DISO value is close to 0, the distance between the observed and simulated values is the closest, and the accuracy is the highest.

$$R = \frac{\sum_{i=1}^{n}(S_i - \bar{S})(O_i - \bar{O})}{\sqrt{\sum_{i=1}^{n}(S_i - \bar{S})^2}\sqrt{\sum_{i=1}^{n}(O_i - \bar{O})^2}} \tag{10}$$

$$AE = \frac{1}{n}\sum_{i=1}^{n}(S_i - O_i) \tag{11}$$

$$RMSE = \sqrt{\frac{1}{n}\sum_{i=1}^{n}(S_i - O_i)^2} \tag{12}$$



$$DISO = \sqrt{(R-1)^2 + \frac{AE^2}{\bar{O}} + \frac{RMSE^2}{\bar{O}}},$$

(13)

where $n$ represents the total number of pixels, $S_i$ and $O_i$ denote the simulated and observed variables at the $i$th pixel, and $\bar{S}$ and $\bar{O}$ are averaged values.

## 4 Results

### 4.1 Adaptive surrogate modeling performance of selected samples

#### 4.1.1 Parameter behaviors

After parameter calibration of all samples, the posterior parameter distributions of the 10 sensitive parameters under the joint constraint of GLASS LAI and GPP were obtained by considering the last 100,000 values. Here, we only showed the posterior distributions of 10 representative pixels that were randomly selected (Fig. 2). Parameter behaviors showing an obvious unimodal pattern were considered as well constrained, the ranges of which were within the previous definition. Most sensitive parameters (except alpha3, theta3, and beta3) exhibited a well-constrained distribution but with different degrees of concentration. GPP had weak constraints on alpha3, theta3, and beta3, mainly reflected in the scattered distributions and wide fluctuation ranges. With the Morris index shown in Fig. 1, we found that GPP seemed insensitive to small perturbations of these parameters. Comparatively, one significant limitation occurred with vmax, the maximum carboxylase capacity of Rubisco, which had a pronounced effect on plant photosynthesis.

The model structure also affected the limiting effect of observations on parameters. The relationship between LAI and its sensitive parameters was more direct without repeated transmission through complex processes. Thus, the posterior distributions of most samples were well constrained. Edge-hitting distribution appeared when the previous ranges of parameters were unsuitable for selected samples, showing that retrieved values were clustered near the highest or lowest bounds of the prior ranges (Fig. 2, 1-3, and 6-6). The rationality of the parameter prior range is not discussed in this paper, but edge-hitting distribution emphasized the importance of prior knowledge for optimization. The differences between samples were related to the spatial variation of parameter sensitivity, as well as the quality of the GLASS GPP and LAI products.

The posterior mean was chosen as the best estimate of each parameter. Thus, we obtained the histogram distribution of the optimal parameters for all samples (Fig. 3). The optimal values of each parameter among samples had noticeable spatial variations that were significantly different from their default values. The Rubisco enzymes exhibited stronger carboxylase capacity, shown as higher vmax after calibration (Fig. 3d), and the leaf turnover rate (p5) also decreased by more than 30% for most samples. For p14, the optimized value was less than half the default value, which indicated that the proportion of carbon in leaf loss transferred to litter was notably reduced. The results proved that using site-scale optimal parameters to define regional optimal schemes could not fully capture the discrepancy of ecophysical characteristics within a PFT, especially when a PFT type covers a large area.





### 4.1.2 Model improvement

With the constrained parameters resulting from the MASM optimization, we took the mean of each posterior distribution as the optimal value and re-entered it into the model to obtain the daily LAI and GPP simulations. Fig. 4 shows the accuracy verification of LAI and GPP simulations against the GLASS products on the scale of every 8 days before and after parameter
optimization.

We calculated the DISO value, a comprehensive indicator used for accuracy verification, for each sample. The histogram of DISO for all samples is illustrated in Fig. 4. For LAI, the DISO of more than 98% pixels was less than 0.6, and the overall mean value was 0.33. For GPP, the DISO of more than 96% of the pixels was less than 0.4, and the overall mean was 0.27. The calibrated IBIS model simulated LAI well on an 8-day basis ($R^2 = 0.85$; RMSE = 0.664 gC m$^{-2}$ 8d$^{-1}$) in calibration years
(2001-2009), while the accuracy of the validation years (2010-2018) slightly decreased. In comparison with GLASS GPP, the overall $R^2$ of the modeled GPP was 0.90, and the RMSE was close to 1.0 gC m$^{-2}$ 8d$^{-1}$. There was no significant difference in accuracy between the calibration and validation years, indicating that the calibrated parameters were stable over time. Our results demonstrated that the MASM approach was effective and efficient for providing approximately optimal parameters for a highly complex terrestrial biosphere model. Most optimization results showed good consistency with GLASS products.

### 4.2 Parameter distribution obtained by the XGBoost approach

### 4.2.1 Feature selection

The parameters calibrated in this study were divided into two categories: (1) parameters related to the photosynthesis process: alpha3, theta3, beta3, and vmax; and (2) parameters related to carbon conversion: p5, p14, p15, p17, p18, and p20 (Table 2). Considering the environmental heterogeneity of each pixel, we explored the relationship between these 10 parameters and
local environmental characteristics from the aspects of climate, soil, location, and important surface variables (like LAI, GPP, and photosynthetically active radiation (PAR) that closely related to photosynthesis) (Table S2). The vapor pressure deficit (VPD), day length, and shortwave radiation data (SRAD) were also considered as influence factors. VPD affects the rate and intensity of evapotranspiration. When plants close their stomata to adapt to high VPD, photosynthesis slows down, and the growth rate slows down (Li et al., 2018b). Shortwave radiation provides the energy for photosynthesis; temperature and
sunshine length in the growing season also affect the gas exchange characteristics of leaves (Rogers, 2014). We also added elevation information as areas with high altitudes are less affected by human activities and may have a higher carbon sequestration capacity compared with low-altitude areas.

### 4.2.2 XGBoost setting

We used the annual mean values of the selected features and the growth season statistics of temperature, VPD, SRAD and
PAR as the inputs of XGBoost modeling. We set 10 optimal parameters obtained by MASM as the target variables (i.e., the outputs of XGBoost). However, it was not guaranteed that posterior values obtained through the MASM approach were the





best selection for each pixel due to observation (GLASS products) quality and algorithm uncertainty. Large errors indicated that calibrated parameters were not suitable for those samples, which would affect spatial prediction if they were included in training data for machine learning. In this paper, the screening indexes were defined according to the DISO values of LAI and

GPP. All the samples with DISO < 0.35 (Fig. 4) were selected as the ML inputs, considering both the accuracy and sizes of the training samples. Finally, a total of 1930 samples were selected for spatial parameter prediction. We used 70% of the selected samples to train the XGBoost model, and the rest as an independent testing set.

### 4.2.3 XGBoost validation

Figure 5 shows the validation after running XGBoost methods. For parameters with strong effects (e.g., vmax, p18) on GPP

and LAI, the fitting accuracy could reach more than 0.8, while parameters with lower sensitivity (like alpha3, theta3, beta3, p14, and p15) tended to have slightly lower accuracy. This is because, in the MASM optimization process, the optimal parameters were mainly constrained by LAI and GPP data, which also led to a more prominent order of LAI and GPP in feature importance ranking when the XGBoost model was trained.

When the parameters from the XGBoost simulation and MASM posterior distributions were used for the IBIS model, the

estimated LAI and GPP showed high correlations with those of the GLASS products (2001–2018). For LAI, the correlation was above 0.6, while that of GPP was mostly above 0.8. Parameters obtained by XGBoost and MASM were highly consistent in capturing the degree of correlation with GLASS products. For the testing set, the estimated errors (RMSE and DISO) using XGBoost were slightly less correlated with the corresponding accuracy indexes of the MASM approach, but the range was similar. DISO was distributed below 0.5 of LAI, while GPP was distributed below 0.3. For the RMSE and DISO indexes,

samples above the diagonal indicated results better estimated using MASM optimized parameters, and those below the diagonal were better estimated by using parameters for the XGBoost simulation. The XGBoost performance was superior to that of MASM in terms of the final validation of parameters. For example, for testing LAI (Fig. 6, c-3), 52% of the pixels showed that the parameters obtained by XGBoost were more accurate, while the ratio reached 60% for testing GPP (Fig. 6, d-3). The possible explanation was that XGBoost used more environment variables related to the parameter; hence, the results

may be more appropriate for each pixel. Moreover, the uncertainty of the MASM algorithm and the diversity of the posterior probability distribution model also affected the selection of the optimal parameters. In addition to better accuracy, another benefit of using the XGBoost method was that the calculation cost of parameter calibration was greatly reduced compared to surrogate modeling optimization.

### 4.2.4 Spatial parameter distribution

By combining the environmental characteristics, the optimal parameter distribution of each pixel was estimated for the entire deciduous forest area (0.05°) in the eastern United States with the application of the trained XGBoost. Fig. 7 shows the spatial distribution of each parameter. Vmax was higher in the central and northeastern parts of the deciduous forests, indicating that the Rubscio enzyme had a higher maximum carboxylase capacity in these regions. Parameters related to vegetation carbon





turnover and allocation include p5, p14, and p15, which indicated a high turnover and allocation ratio in the western and
northwestern areas. Leaf mass per area, shown as p18, was low in the northeast and about three times higher in parts of the
South and Southwest of the study region.

Compared with the original parameter ranges, the posterior distributions were significantly more concentrated (see
'Pars_Hist' in Fig. 7). The overall ranges were greatly reduced, especially for theta3, beta3, p15, p17, and p18. There was a
great difference in the distribution between optimal parameters and default values in the whole study area. For theta3 and beta3,
the default values were outside the whole statistical range, which would induce a large error when using defaults to estimate
carbon fluxes. For vmax, the optimal values were higher than the default values and had evident spatial heterogeneity. Vmax
is a key source of uncertainty in current ecosystem models, and adopting a fixed value may give rise to large systematic error
(Croft et al., 2017; Bonan et al., 2011; Liu et al., 2014; Walker et al., 2014). Rogers (2014) surveyed the derivation of vmax
in earth system models and found a wide range of variation among vegetation types, and Bonan et al. (2011) showed that
model uncertainty resulting from this parameter was comparable with that from the model structure.

### 4.3 Model improvement

### 4.3.1 Uncertainties in optimized LAI and GPP

Using the parameters estimated by XGBoost, we obtained the spatiotemporal distribution of optimized LAI and GPP. The
estimated LAI and GPP before and after optimization were compared with those of the GLASS products. Fig. 8 shows the
frequency distribution of the accuracy indexes ($R^2$, RMSE, Bias, and DISO) of pixel-by-pixel validation in the deciduous
forests from 2001 to 2018. The correlation coefficients of LAI and GPP before and after optimization showed little change.
For LAI, the correlation of most pixels was above 0.6 (Fig. 8, a-1), and for GPP, it was above 0.8 (Fig. 8, b-1). However, the
RMSE of the optimized parameters was reduced by 0.56 and 0.45 for LAI and GPP, respectively. We comprehensively
evaluated the performance of both LAI and GPP with the DISO index. We found that for LAI, the DISO was less than 1 for
60% of the pixels before optimization and was less than 0.3 for nearly 90% of the pixels after optimization. Similarly, for GPP,
the DISO decreased by 0.16 on average, and the overall distribution became more concentrated.

Figure 9 manifests the spatial distribution of the absolute error between estimated variables and GLASS products. The error
distributions of the default parameters indicated that the error of LAI and GPP had similar spatial distributions. A high
discrepancy of the previous model compared with GLASS mainly occurred in the middle and northwest parts of the study area.
The absolute errors were dramatically improved after optimization. For LAI, the overall error was below 0.6 $m^2$ $m^{-2}$; for GPP,
the errors in some central areas were between 0.6 gC $m^{-2}$ $8d^{-1}$ and 0.9 gC $m^{-2}$ $8d^{-1}$, and they were below 0.6 gC $m^{-2}$ $8d^{-1}$ for the
rest. The reduction of model errors was not consistent across pixels dominated by deciduous forests.





### 4.3.2 site-level validation in carbon fluxes

Compared with the default estimates, our optimized fluxes had improved with RMSE reduced by 12% for GPP, 20.38% for
ER and 1.57% for NEE, while the correlation coefficients decreased slightly for GPP and NEE (Fig. S2). Using DISO as a comprehensive evaluation indicator, we verified the GPP before and after optimization at 14 flux sites, and also summarised the effect of parameter optimization on ER and NEE for every 8-day. When the GLASS GPP product was regarded as the 'true value', the optimized model successfully estimated the magnitudes and temporal variations of GPP, and the DISO values of several sites improved by more than half throughout the whole year (e.g., US-Bar, US-LPH) (Fig. 10, grey bars). This is
because the GLASS GPP product was used as the reference while calibrating the sensitive parameters. Considering the uncertainties of the GLASS product, we evaluated GPP, ER and NEE using data from flux observation sites as a 'true value'. Collectively, the optimized model improved the flux estimates at most flux towers compared with those from the original model ($\triangle$DISO > 0). We also evaluated the model performances in the growing season and non-growing season, and found that GPP and NEE in the growing season were improved at most of the sites such as US-Ha1, US-UMB, US-Umd and US-
WCr. However, for the non-growing season (NGS), the GPP accuracy decreased, and the corresponding NEE also showed greater inconsistency with the flux site NEE. Larger time-series differences during NGS between optimized and observed fluxes also existed at most sites compared to the default fluxes. The decrease of GPP accuracy is the main contributor to the larger errors of NEE estimates for most flux sites, while for US-Wi3, ER showed lower accuracy.

We calculated the DISO indicators between GLASS GPP and flux observations for the three periods (annual, growing
season, and non-growing season) (see DGLA values shown in Fig. 10). The values of DISO in the non-growing season were significantly higher than those in GS, especially at the US-Bar, US-Wi1 and US-LPH sites, which indicated that GLASS GPP had a lower performance in the non-growing season for these 14 sites. This also explains why the difference between the optimized carbon fluxes and the observation was larger in the non-growing season. For US-Wi8, the GLASS GPP differed greatly from the observed data (with DISO of 1.12), and therefore the optimized results showed lower accuracy than those
based on the default parameters. The timing of the peak GPP generally closely matched that of the flux tower GPP, although there was still significant underestimation for several sites (e.g., US-LPH, US-MMS and US-Oho).

### 4.3.3 Impacts on simulated regional vegetation carbon fluxes

We used our optimized model to estimate GPP, ER and NEE for each 0.05°×0.05° pixel across the deciduous forests in the eastern United States from 2000 to 2019. The spatial patterns of the 20-yr mean annual carbon fluxes are shown in Fig. 11.
For the simulation based on the default parameters, the overall distribution of GPP gradually increased from high latitude to low latitude; a similar trend was also shown in ER; a weak carbon sink was observed for the whole region. The patterns changed obviously after parameter optimization. After the optimization, the Southeast of the US and in particular the areas along the Appalachian Mountains, an ecoregion of temperate broadleaf and mixed forests, had the highest GPP (~1500-2000 gC m$^{-2}$ year$^{-1}$); lower annual GPP was found in the West of the study region; ER generally exhibited similar patterns, and was





systematically lower than GPP estimates. For part of the western region, NEE was less than 200 gC m$^{-2}$ year$^{-1}$, also in correspondence with areas characterized by low estimated GPP and ER. The spatial patterns of carbon fluxes with optimized parameters exhibited obvious spatial details and provided more reasonable spatial information.

Figure 12 showed the interannual variability in carbon fluxes (Fig.12, a) and percentages of variation after optimization for the whole year, the growing season and non-growing season (Fig.12, b-c). The mean annual GPP, ER and NEE over the

deciduous forests in the eastern United States for the period from 2000 to 2019 were 5.79, 4.60 and -1.19 Pg year$^{-1}$, respectively. The optimized GPP and ER were significantly higher than the defaults, and the capacity of carbon sequestration in deciduous forests also increased by more than half. We compared the impacts of the optimized model on different periods (annual, growing season and non-growing season) and found that our estimates of GPP and ER were higher than previous estimates, of which the increase in the non-growing season was the most significant. For NEE, the capacity of deciduous forests to absorb

carbon increased slightly in the growing season, while the estimates of carbon release decreased in the non-growing season, leading to an increase in carbon sequestration throughout the year.

## 5 Discussion

### 5.1 Algorithmic uncertainty

Our pixel-level calibration was divided into two parts. The first part used the MASM algorithm to calibrate each sample. We

recorded the simulated CF of GPR model constructed in each adaptive resampling and the simulated value of the real model, which was expressed by GPR$_{RMSE}$ to measure the accuracy of the GPR model. We also calculated the average of the last 100,000 CFs after finishing the second iteration optimization, which was represented by CF$_{mean}$.

It was found that they were correlated and had unusually high values (Fig. S3). When the error of the GPR model was large (i.e., the GPR model cannot accurately represent the IBIS model), it was difficult to produce a smaller CF through iterative

optimization. Therefore, the optimal parameter scheme for such samples would cause large errors when simulating LAI and GPP. From an algorithmic perspective, the number of iterations may have been insufficient during MCMC sampling. However, it should be noted that the increased number of iterations definitely increased the calculation cost. Two reasons could explain MASM uncertainty: (1) different samples may have had different sensitivity to parameters, and therefore we could not guarantee that the sensitive parameters of every sample were highly correlated with the model error; and (2) the prior range of

parameters may have been inappropriate, failing to obtain a suitable combination. To ensure the accuracy of the input training set during the later training of the XGBoost model, sample pixels with excessive GPR$_{RMSE}$ and CF$_{mean}$ values were eliminated. In addition, most sample pixels converged to a stationary distribution with GR < 1.2 in the second iteration optimization, while very few pixels that did not meet the convergence condition after 10 cycles were eliminated. We also performed quality control on samples when using XGBoost to simulate spatial optimal parameters to ensure the accuracy of input parameters.





## 5.2 Uncertainty of optimal parameters

It was found that the posterior distribution of the optimized parameters in each pixel had significant differences in distribution forms (single peak or multipeak), mean values, and fluctuations. The poorly constrained results meant that the model predictions were slightly sensitive while changing those parameters; for a multipeak distribution, there were multiple combination schemes for this sample that met the requirement of minimum CF; edge patterns representing posterior values were skewed to one side of the previous ranges, which reflected the defect of the model structures or the irrationality of the prior parameter ranges (Liu et al., 2015; Mäkelä et al., 2019). The uncertainty boundaries of these parameters were likely to be unrealistic and could lead to overconfidence in model predictions (Lu et al., 2017). The limiting effect of observations was strongly related to the sensitivity of observed variables to parameters, which indicated that the spatial variability of parameter sensitivity should also be considered in parameter optimization. The interaction between parameters should also be considered in parameter estimation (Fig. S4).

We provided an optimal parameter scheme for each pixel of the eastern United States and a more concentrated range of proposed parameters through the calibration that may be helpful for others to further find optimal parameters with high efficiency within this area. Although there had been many studies on parameter calibration, significant inconsistency still existed among the same parameters for different ecosystem models. This was mainly because the model and input data errors were compensated by parameter adjustment, and therefore it was difficult to ensure that the estimated parameters could be explained theoretically. When expanding the spatial domain of parameters, the limited understanding of the influences on each parameter also prevented us from estimating the actual values of these parameters. For example, previous studies found that vmax is closely related to leaf nitrogen content, and an increase of phosphorus content in leaves significantly improves the sensitivity of vmax to leaf nitrogen (Walker et al., 2014). Leaf and labile carbon turnover rates (p5, p15) were key factors determining the carbon sequestration capacity of terrestrial ecosystems. Wang et al. (2017) studied the biological and abiotic factors affecting forest carbon turnover time through quadrat observation and showed that multiple factors such as soil nutrients (e.g., carbon, nitrogen, and phosphorus), pH, and forest ages could not be ignored. It was also pointed out that the carbon turnover time could vary with time, which was not considered in this study. In addition, the calculation of LAI in DALEC was so simple that the sensitive parameters involved were correlated with LAI values. However, the calculation of GPP in IBIS involved many complex biochemical processes and factors, and thus the effect of GPP on parameters was not apparent. We could not guarantee that the parameters of each pixel were well-constrained in sample calibration. For samples with poor and edge-hitting constraints, if the simulated LAI and GPP showed good accuracy, we also took the optimal values as training samples for XGBoost, which could negatively impact model prediction.

## 5.3 The quality of optimized carbon fluxes

Generally, the previously estimated carbon fluxes matched the distribution of several key environmental variables (e.g., temperature, day length, radiation, specific humidity) more closely (Fig. S5), the trend of which was that the photosynthesis





and respiration of deciduous forests gradually increased with the decrease of latitude. The information provided by climate conditions was reflected in the carbon flux simulation through the simulation of the model. As the physiological parameters used in each pixel were uniform values in the original model, and only a single vegetation type was considered, the sources of spatial differences in carbon fluxes were mainly from the differences in driving data sets. The integration of GLASS products introduced the spatial distribution pattern information of the LAI and GPP products into the model by adjusting the distribution pattern of key parameters, which made the optimized spatial pattern distribution more reasonable. The optimized model with calibrated parameters can provide more accurate LAI and GPP information and the temporal and spatial distribution of both LAI and GPP were closer to those of the GLASS products. Although a decrease in the RMSE of the optimized fluxes performed a better validation, particularly around the period of peak fluxes, the results also indicated that when there is quite a distinction between GLASS and ground observation, it is difficult to successfully capture the variations of each flux site. For example, our results of most sites (e.g., US-Bar, US-WCr and US-Wi3) showed that optimized NEE exhibited overestimation of net carbon uptake during the non-growing season.

It is already known that the accuracy of spatial reference products (GLASS products in this paper) is the key factor affecting the accuracy of carbon fluxes during the model-data fusion. Multiple data streams and more relevant state variables should be applied as a way to mitigate the deviations from a single product or variable. We considered both LAI and GPP in the optimization, but their contributions to the model improvement were not evaluated separately, which should be taken into account in future work. ML provides a convenient approach for integrating spatial products with physical models, and we expect more explorations on developing a hybrid modeling framework to couple ML with physical models or explain and even compensate model errors due to lack of prior knowledge. Such a combination can increase the credibility of future carbon budget estimation and strengthen the rationality and interpretability of ML. In addition, we also put forward a demand for higher resolution and high-precision remote sensing products, which are necessary for model improvement and for carrying out benchmarking tests to comprehensively evaluate the performances of different models.

## 6 Conclusions

It was generally accepted that the model parameters had spatial heterogeneity, but calibrating a complex process-based model at the pixel level was not realistic, especially with an increase in spatial resolution. This paper proposed a two-step framework for estimating optimal parameters at the pixel level. We randomly sampled the study area and used the GPR model as the surrogate model, and then applied the MASM algorithm for iterative optimization to obtain the posterior distributions of samples. Next, we used XGBoost to describe the nonlinear relationship between optimal parameters and local climate, soil, and surface variables and extend to the entire deciduous forests in the eastern United States. Our method provided an optimal parameter scheme for each pixel and confirmed that the discrepancy between GLASS products and predicted values was significantly reduced with the optimized parameters. The results showed that there was significant spatial variability of parameters within a vegetation type, and that using high-quality remote sensing products could efficiently calibrate the

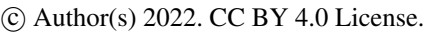

parameters of terrestrial biosphere models at the pixel level. Although we tested our approach only for deciduous forests of
510 the eastern United States, it provided a feasible scheme for spatial calibration of other vegetation types, at higher resolutions
and in larger areas.

*Code and data availability*. The original IBIS model realized in the fortran language has been released in its 2.5 version. It is
openly shared without restriction, and can be obtained freely from https://daac.ornl.gov/cgi-bin/dsviewer.pl?ds_id=808. The
DALEC (version2) model code is a publicly released version from https://datashare.ed.ac.uk/handle/10283/3267 (Williams,
2019). The standalone version of ASMO-PODE can be accessed directly from https://github.com/gongw03/ASMO-PODE
(Gong et al., 2017). We provide an optimization script of one pixel as well as its driving data as material to support the
implementation of MASM algorithm in calibrating the IBIS simulator. All source code and the additional information of main
codes used in our manuscript are packaged at https://github.com/rmars6922/MASM. For any questions or interest in our model
data or code, please contact the corresponding authors.

*Author contribution*. RM, SL, and JX conceived the study. RM performed the analysis. RM, SL, and JX interpreted the results.
RM wrote the draft. JX, HM and TH revised the manuscript. DG and XL contributed to the preprocessing of all the model
drivings, and helped with running machine learning approach. HL guided the operation of IBIS model. All authors contributed
to the preparation of the manuscript.

*Competing interests*. The authors declare that they have no conflict of interest.

*Acknowledgements*. This study was partially supported by the National Key Research and Development Program of China
(NO.2016YFA0600103), and the National Natural Science Foundation of China (No. 42090011). JX was supported by the
University of New Hampshire. We gratefully acknowledge the data support from "National Earth System Science Data Center,
National Science & Technology Infrastructure of China. (http://www.geodata.cn)". We thank the PIs of the FLUXNET2015
Tier-1 sites and AmeriFlux network for contributions to the flux measurements in this study. We thank Wei Gong for providing
detailed guidance on the ASMO-PODE approach.

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






**Tables and Figures**

**Table 1.** Description of the datasets used in this study[a]

|  | Products | Time spin | Spatial Res. | Temporal Res. | Variables | Reference |
|---|---|---|---|---|---|---|
| Category 1: Model Forcing (model inputs) | | | | | | |
| Climate | Daymet | 2000-2019 | 1km | Daily | tmin, tmax, prec | Thornton et al. (2021) |
|  | GRIDMET | 2000-2019 | 2.5′ | Daily | vs, sph, | Abatzoglou (2013) |
|  | NLDAS-2 | 2000-2019 | 0.125° | Hourly | pa | LDAS (2016) |
|  | NARR | 2000-2019 | 32km | Daily | cld | NCEP (2005) |
| Auxiliary | $CO_2$ concentration | 2000-2019 | - | Yearly | $CO_2$ | Thoning et al. (1989) |
|  | GSDE | - | 30″ | - | soil sand, soil clay | Shangguan et al. (2014) |
|  | NLCD | 2001 | 30m | Yearly | Land cover | Homer et al. (2020) |
| Category 2: Ground Observations (NEE validation) | | | | | | |
|  | FluxNet2015 | Site | Site | Daily | NEE | Pastorello et al. (2020) |
|  | AmeriFlux | Site | Site | Daily | NEE | - |
| Category 3: Satellite Products (MDF and validation) | | | | | | |
|  | GLASS LAI | 2000-2015 | 0.05° | 8-day | LAI | Ma et al. (2021) |
|  | GLASS GPP | 2000-2017 | 0.05° | 8-day | GPP | Zheng et al. (2020) |

[a] Abbreviations: Res: Resolution; tmax: the maximum air temperature (°C); tmin: the minimum air temperature (°C); prec: precipitation (mm); vs: wind speed (ms[-1]); sph: specific humidity (%); pa: pressure (Pa); cld: cloud fraction (%)

**Table 2**. Description and prior ranges of IBIS-DALEC parameters

| No. | Parameters | Description | Unit | Default | Min | Max |
|---|---|---|---|---|---|---|
| Parameters for IBIS model | | | | | | |
| 1 | alpha3 | intrinsic quantum efficiency for c3 plants | - | 0.06 | 0.05 | 0.09 |
| 2 | theta3 | coupling coefficient of C3 photosynthesis |  | 0.97 | 0.6 | 0.996 |
| 3 | beta3 | coupling coefficient of C3 photosynthesis | - | 0.99 | 0.6 | 0.996 |
| 4 | gammaub | intrinsic quantum efficiency for c3 |  | 0.015 | 0.0075 | 0.0225 |
| 5 | coefmub | 'm' coefficients for stomatal conductance relationship |  | 10 | 8 | 15 |
| 6 | coefbub | 'b' coefficients for stomatal conductance relationship |  | 0.01 | 0.005 | 0.015 |
| 7 | vmax | the maximum carboxylase capacity of Rubisco | $molCO_2\,m^{-2}\,s^{-1}$ | 3.00E-05 | 2.00E-05 | 8.00E-05 |





| 8 | specla | specific leaf area | $m^2\,kg^{-1}$ | 12.5 | 10 | 20 |
|---|---|---|---|---|---|---|
| 9 | tauleaf | foliar biomass turnover time constant | yr | 2 | 0.05 | 5 |
| 10 | tauroot | fine root biomass turnover time constant | yr | 1 | 0.05 | 1.5 |
| 11 | tauwood0 | stem biomass turnover time constant | yr | 50 | 25 | 75 |
| 12 | aleaf | carbon allocation fraction to leaves | | 0.3 | 0.15 | 0.45 |
| 13 | aroot | carbon allocation fraction to fine roots | | 0.2 | 0.05 | 0.35 |
| 14 | tempvm | parameter thermal stress Vmax | - | 3500 | 3000 | 4000 |
| 15 | rgrowth | growth respiration coefficient | - | 0.3 | 0.1 | 0.5 |
| 16 | rroot | maintenance respiration coefficient for root | $s^{-1}$ | 1.25 | 0.625 | 2.5 |
| 17 | rwood | maintenance respiration coefficient for wood | $s^{-1}$ | 0.0125 | 0.00625 | 0.2 |
| 18 | beta1 | parameter related to the distribution of fine root lower | - | 0.95 | 0.75 | 0.99 |
| 19 | beta2 | parameter related to the distribution of fine root upper | - | 0.975 | 0.75 | 0.99 |
| 20 | stressfac | moisture stress factor | - | -5 | -6.5 | -3.5 |
| Parameters for DALEC module | | | | | | |
| 21 | p2 | Fraction of GPP respired autotrophically | | 0.45 | 0.2 | 0.7 |
| 22 | p3 | Fraction of NPP allocated to foliage | | 0.4 | 0.01 | 0.5 |
| 23 | p5 | Turnover rate of foliage | $d^{-1}$ | 0.06 | 1.00E-04 | 0.1 |
| 24 | p10 | Parameter in temperature sensitivite rate | | 0.073 | 0.05 | 0.2 |
| 25 | p12 | GDD value causing leaf out | | 240 | 200 | 400 |
| 26 | p13 | Minimum daily temperature causing leaf fall | | 9 | 8 | 15 |
| 27 | p14 | Fraction of C in leaf loss transferred to litter | | 0.48 | 0.1 | 0.7 |
| 28 | p15 | Turnover rate of labile carbon | $d^{-1}$ | 0.09 | 1.00E-04 | 0.2 |
| 29 | p16 | Fraction of labile transfers respired | | 0.15 | 0.01 | 0.5 |
| 30 | p17 | Maximum Cf value | $gC\,m^{-2}$ | 300 | 100 | 500 |
| 31 | p18 | Leaf mass per area (lma) | $gC\,m^{-2}$ | 60 | 10 | 200 |
| 32 | p19 | threshold | | 100 | 80 | 120 |
| 33 | p20 | threshold | | 270 | 200 | 365 |






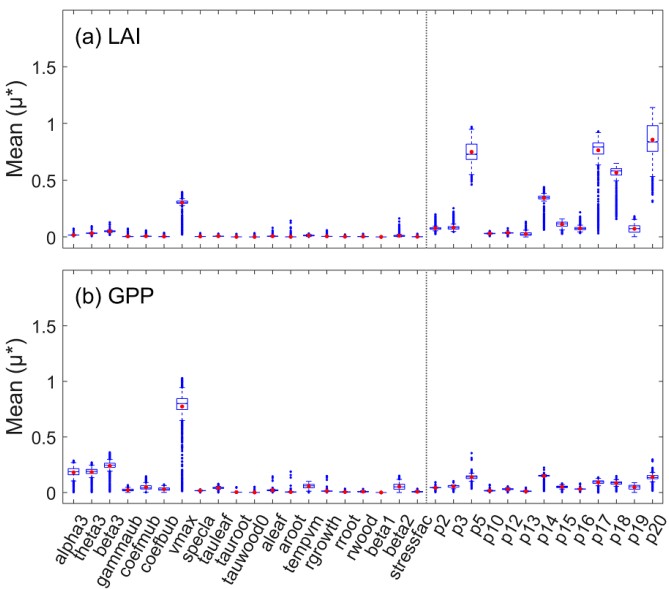

**Figure 1**. Sensitivity of (a) LAI and (b) GPP to all the parameters. The box chart shows the average and quartile of the Morris sensitivity index for all the samples. The red dots represent the mean, the blue midline means the median, the black bottom and top lines of the rectangle are the maximum and minimum values, and the blue dots denotes the outliers.



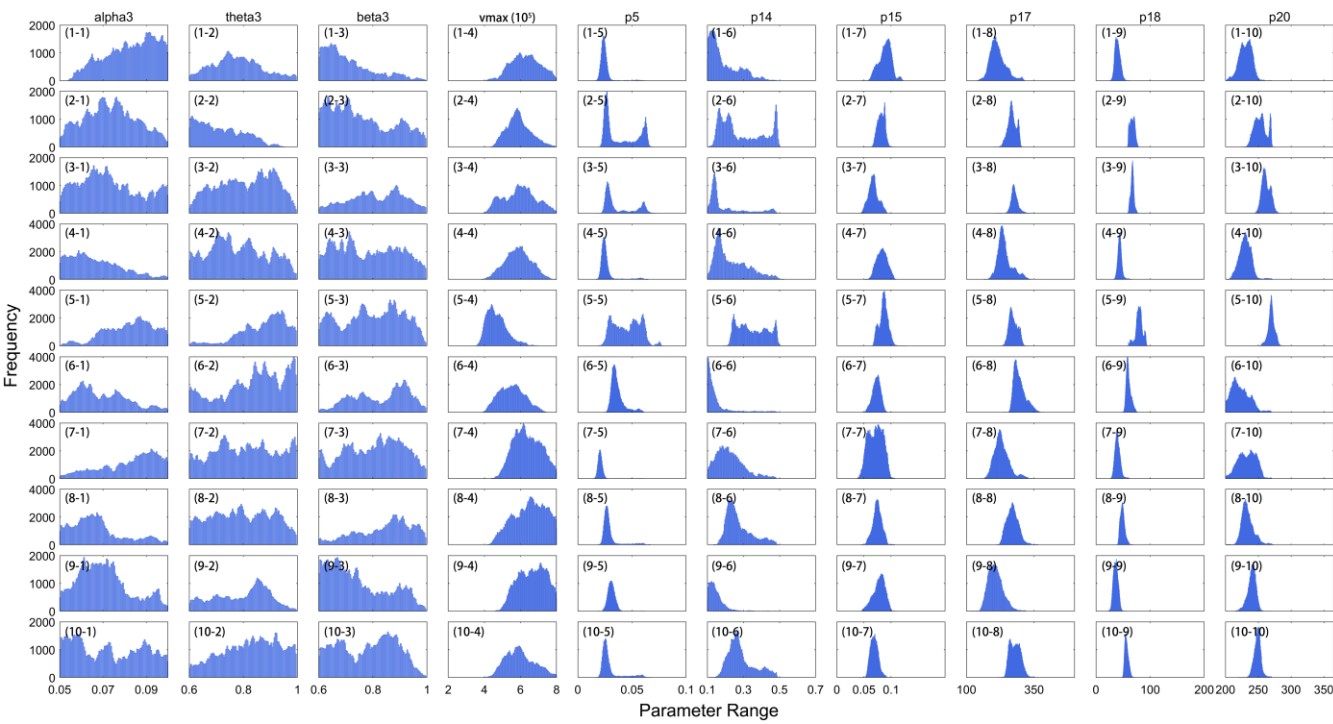

**Figure 2.** Posterior probability distribution of ten representative samples.

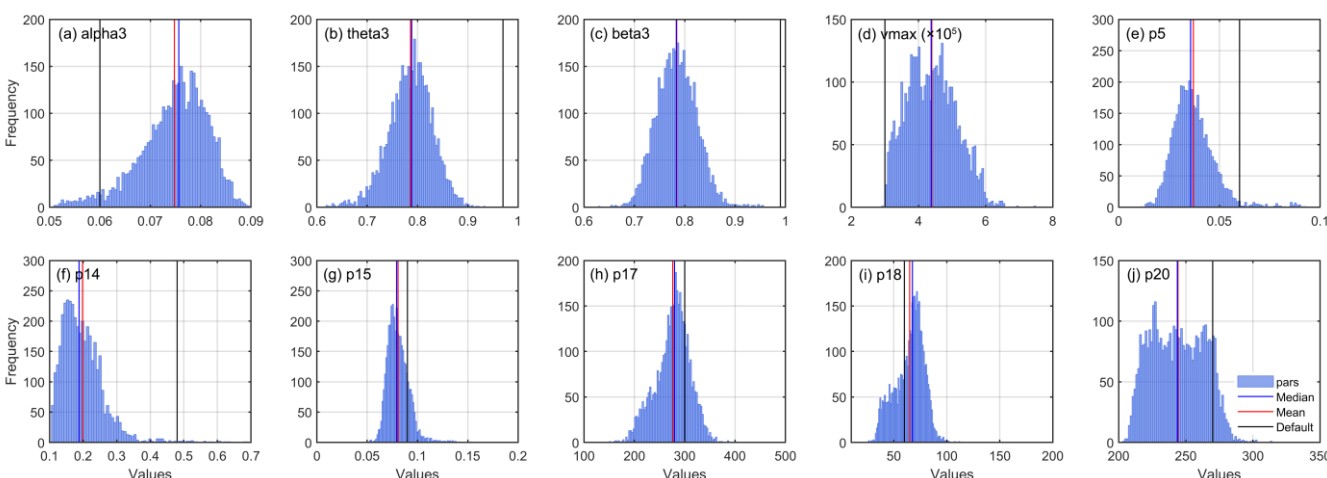

**Figure 3.** Histogram distribution of optimal parameters from all samples using the MASM optimization.



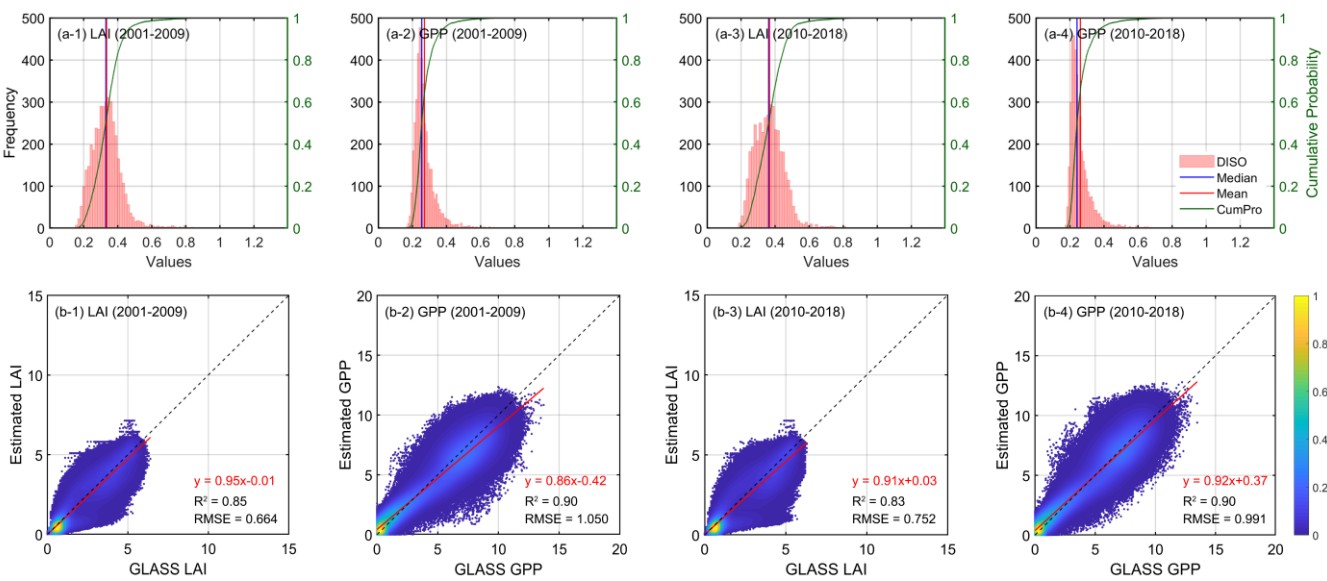


**Figure 4.** Validation of samples with optimal parameter schemes: (a) histogram statistics of DISO index for LAI (a-1, a-2) and GPP (a-3, a-4), and scatter density plot of LAI (b-1, b-2) and GPP (b-3, b-4) against GLASS products on eight-day basis. Some samples that do not converge during the MASM process and show anomalous high errors are excluded. The calibration years are from 2001 to 2009, while the validation is from 2010 to 2018. 'CumPro' in (a) represent the cumulative probability.

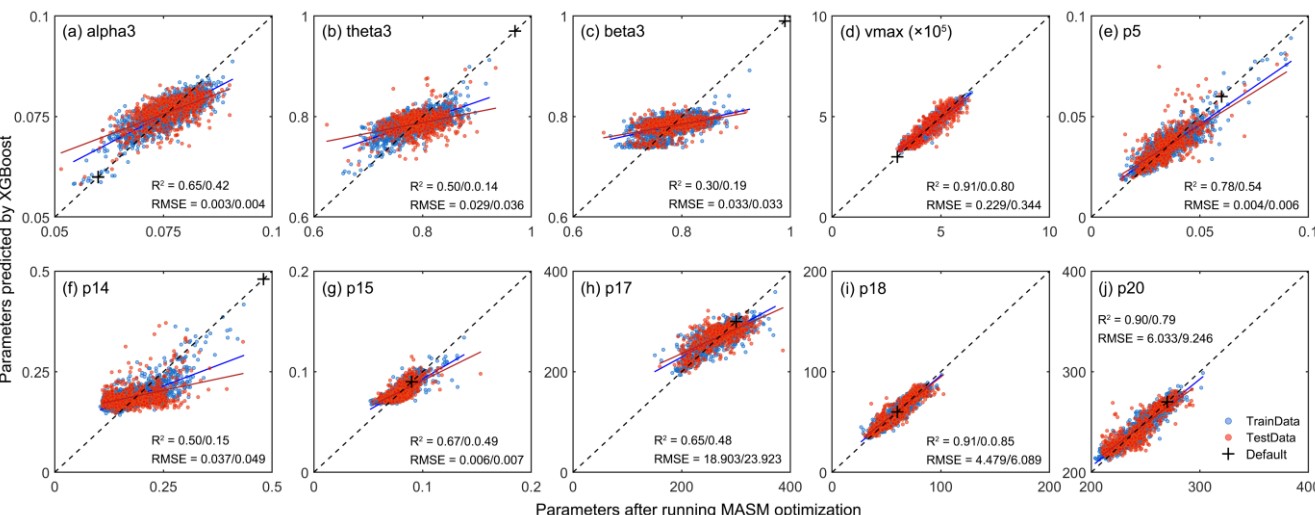


**Figure 5.** Scatter plots of the results of the XGBoost model. The X-axis represents parameters after running MASM optimization, Y-axis means values obtained from XGBoost model. The means before and after the slash represent the accuracy indexes of the training set and testing sets, respectively. The blue dots represent the training set, and red dots represent the testing set. The black cross indicates default values.

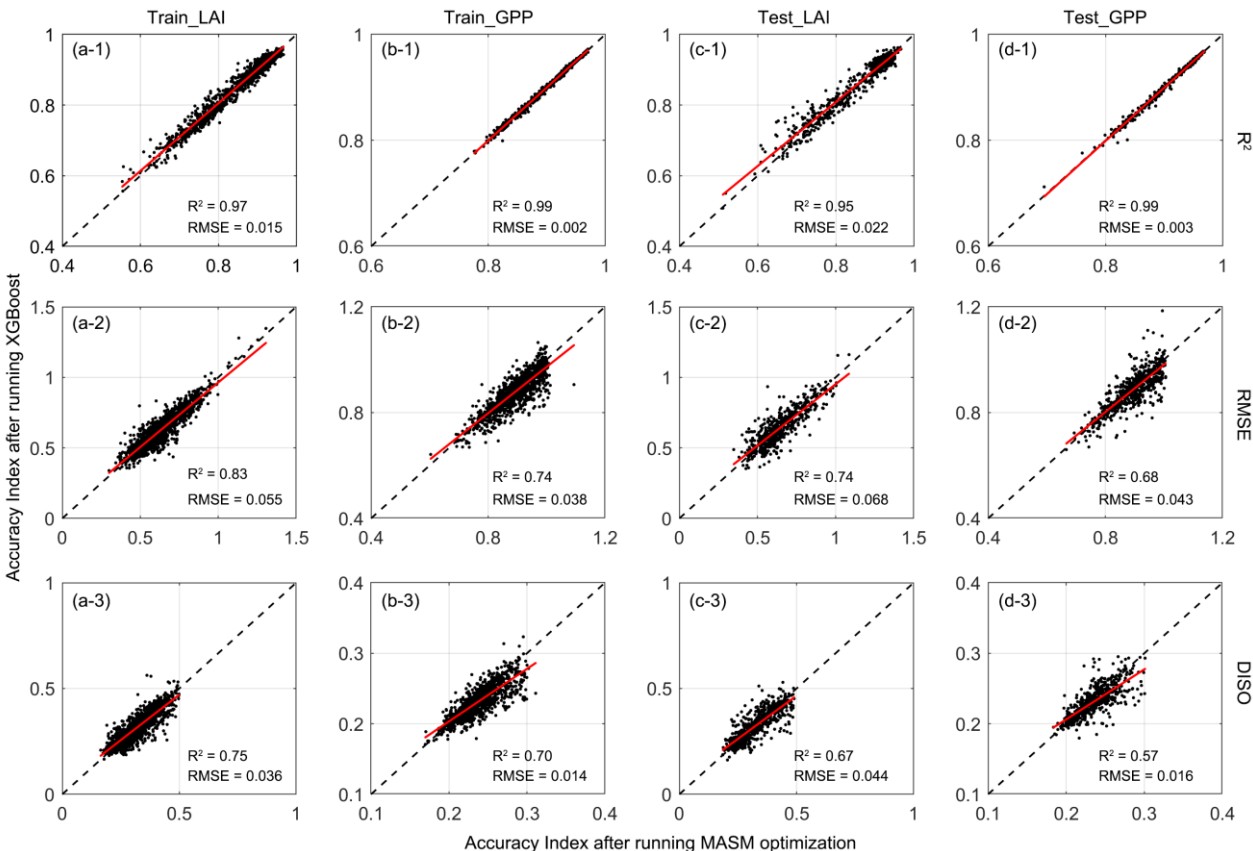

**Figure 6.** Accuracy indexes (R², RMSE, and DISO) validation between trained and predicted parameters in simulating LAI and GPP. The X-axis represents the accuracy index using MASM posterior parameters; The Y-axis represents the accuracy index using estimated parameters from the trained XGBoost model.

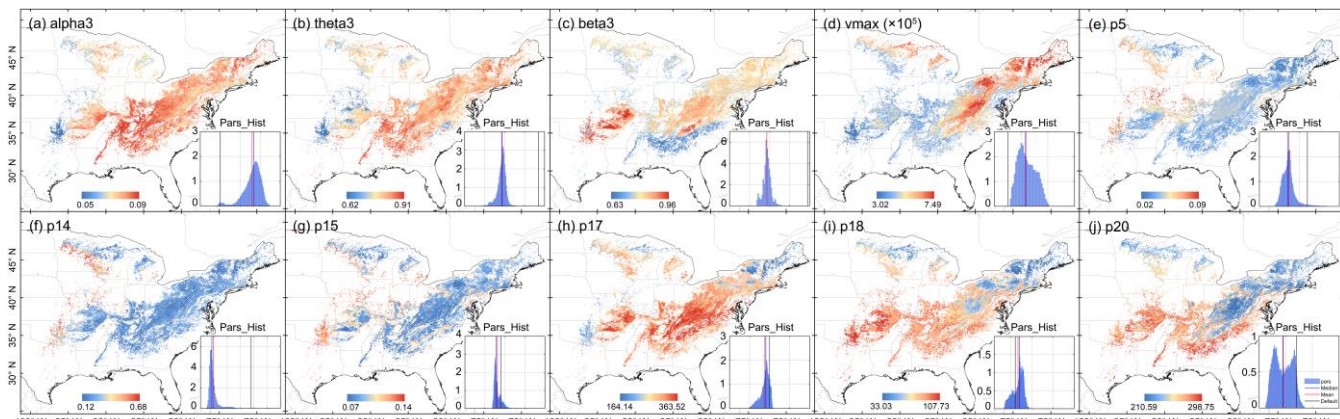

**Figure 7.** Spatial distribution of posterior parameter**.** The inset in the lower right corner of each plot represents the spatial distribution histogram of each parameter



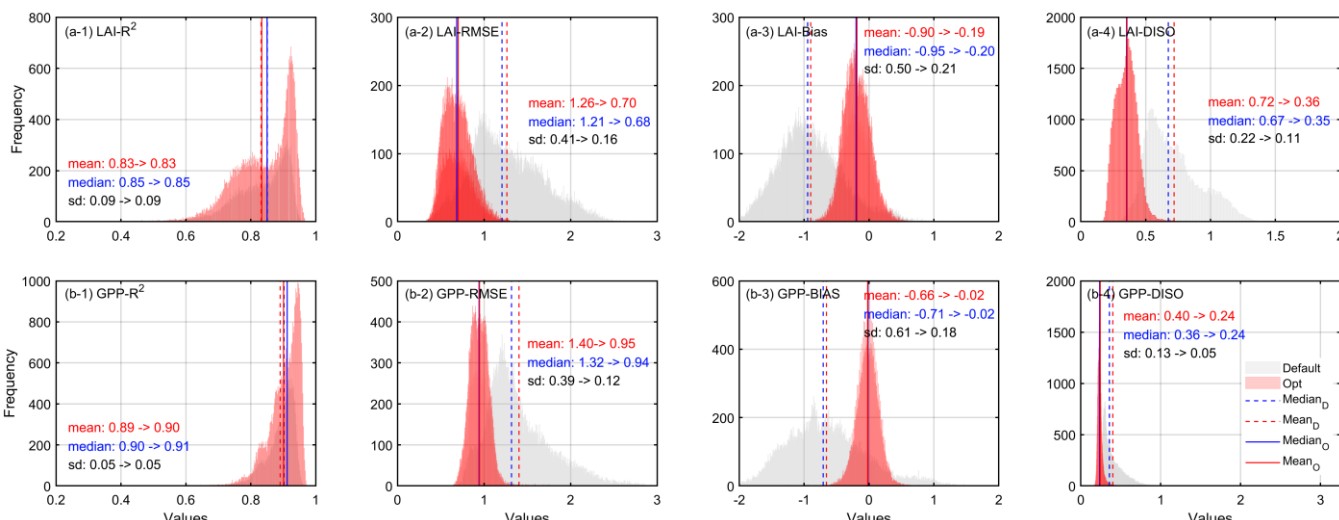

**Figure 8.** Histogram error statistics of LAI and GPP by running prior and posterior parameter schemes.

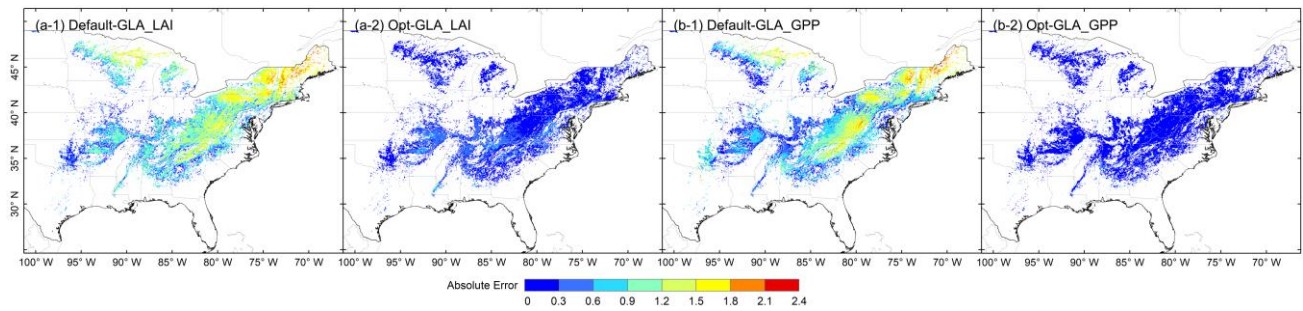


**Figure 9.** Spatial distribution of annual absolute error between model results and GLASS products from 2001 to 2018.: (a-1) and (a-2) are the absolute difference of prior and posterior LAI, respectively; (b-1) and (b-2) are the absolute difference of prior and posterior GPP, respectively.



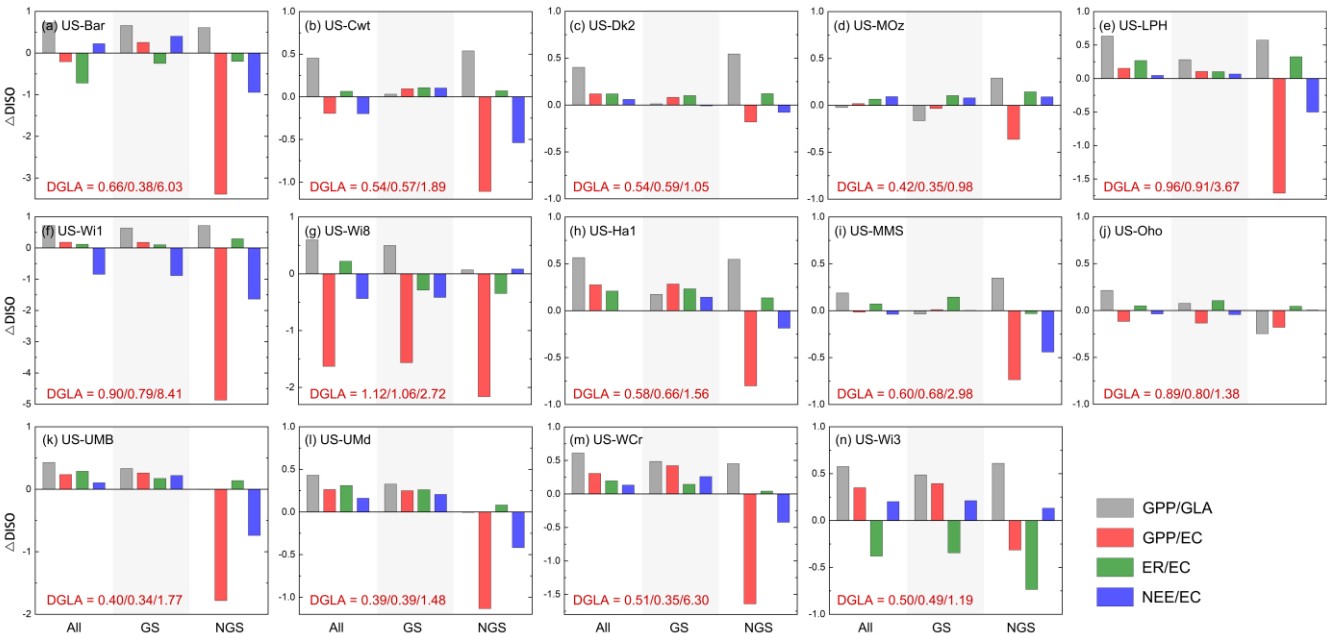

**Figure 10**. Optimized carbon fluxes performances for 14 flux sites throughout the whole year (All), growing season (GS, May to September) and non-growing season periods (NGS). The y-axis means the changes between the DISO indexes before and after the optimization: $\triangle$DISO $= - (DISO_{Opt} - DISO_{Def})/DISO_{Def}$. Positive values represent improvement. The grey bars (GPP/GLA) represent the DISO index between the modelled GPP and the GLASS GPP products, while the red (GPP/EC), green (ER/EC) and blue (NEE/EC) bars mean the DISO index between the modelled fluxes (GPP, ER and NEE) and the observed values at each flux site, respectively. EC represents the flux observations. The bottom left of each pattern shows the DISO index between the GLASS GPP product and the observations throughout three periods (All, GS and NGS).



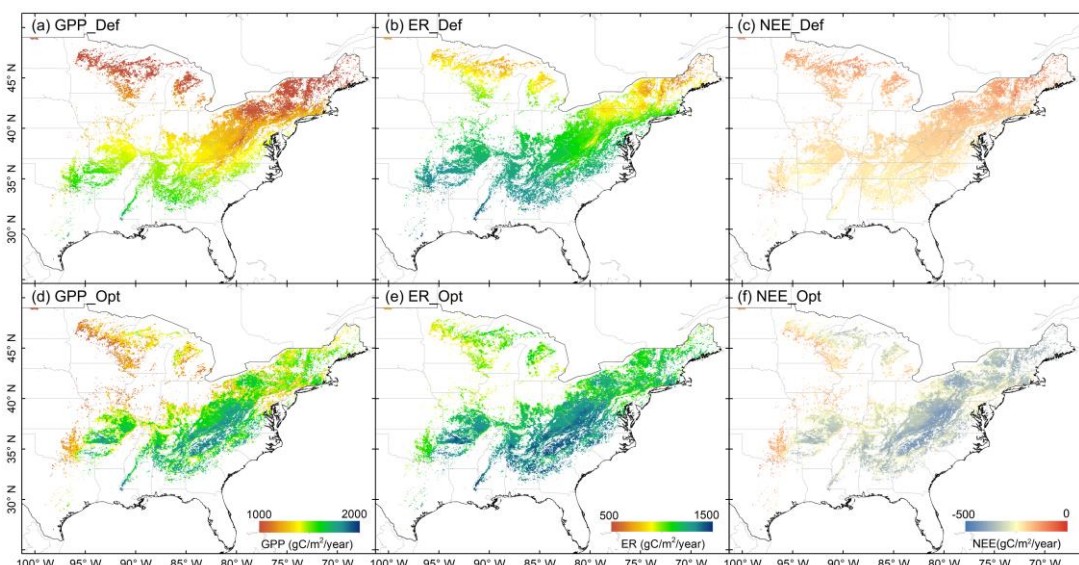

**Figure 11** Regional patterns of mean annual carbon fluxes (GPP, NEE and ER) for the years 2000-2019. Panels (a-c) show spatial distribution
of each flux from the model with default parameters, (d-f) show distributions with optimized scheme.

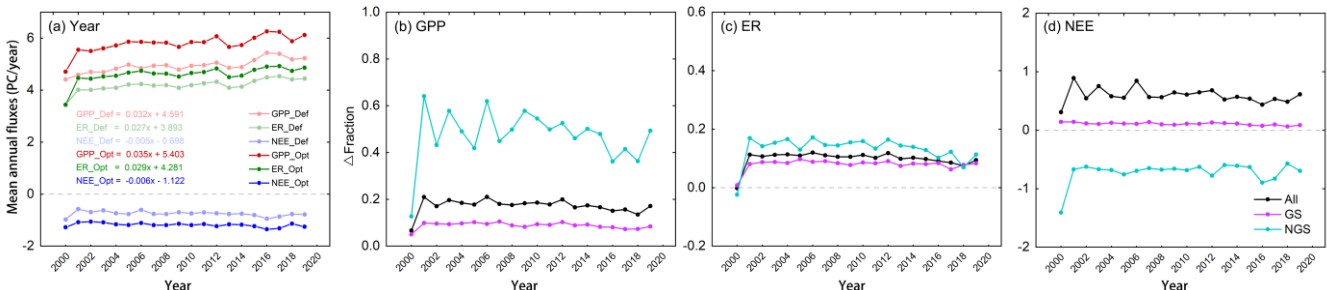

**Figure 12.** Temporal variations of each carbon flux (GPP, NEE and ER) over the deciduous forests in the eastern U.S. (2000-2019). (a)
shows the interannual variation before and after parameter optimization. (b-c) represent the fraction changes after optimization, where
positive fraction indicates an increase from the default fluxes, negative value indicates a decreasing impact.