# Peer review of "Pixel-level parameter optimization of a terrestrial biosphere model for improving estimation of carbon fluxes with an efficient model-data fusion method and satellite-derived LAI and GPP data"

_Geoscientific Model Development, 2022_

## Author Comment (AC5)

**Reviewer2**

This paper proposed a two-step framework to estimate the essential parameters of the revised Integrated Biosphere Simulator (IBIS) at pixel level. The paper was well prepared, organized and clearly presented. However, there are still some issues that the authors should address before the paper is considered for the publication.

Thank you very much for your comments. We provide point-to-point responses to your comments.

1. This study used global LAI and GPP products from the Global Land Surface Satellite (GLASS) suite as "observations" ("true values") for parameter calibration on a spatial scale. If the GLASS product is good enough to serve as "true values", what is the significance of model parameter correction? Please clarify it clearly in Introduction.

**Response:**

Using remote sensing products as a reference to calibrate the parameters of the surface model can not only improve the simulation accuracy of our target variables (like GPP and LAI in our manuscript) but also affect the entire terrestrial ecosystem processes of the model, which can thereby improve the simulation of other related variables (such as NEE, respiration, etc.), and our understanding of the terrestrial ecosystem processes and their interaction with the environment.

We have clarified this point in our introduction section, please check line 56-60.

2. The paper mainly focused on deciduous forests in the eastern United States. How about the accuracy of other products in the study area? If you want to take GLASS products as the reference values, please elaborate the accuracy of GLASS products used in the study area.

**Response:**

The GLASS LAI and GPP products have been validated against globally available ground measurements, and compared with several different simulations from other products. The results showed that GLASS products performed well in accuracy validation and spatial-temporal variations (Zheng et al., 2020; Ma and Liang, 2022). We also compared the accuracy differences between GLASS GPP product and several other GPP products (BESS, VPM, FLUXCOM, and MODIS) in our research region, and found that GLASS estimated GPP fairly well. The $R^2$ of GLASS is almost identical with that of other products; the RMSE of GLASS is slightly lower than that of BESS and comparable to that of VPM, FLUXCOM, and MODIS.

[Figure]

Fig. Accuracy comparison among different GPP products for 14 flux sites.

In fact, no matter which set of products is used, there will be certain errors in the product itself. The focus of this paper is to emphasize that the two-step optimization algorithm that we developed can obtain the optimal parameter scheme pixel by pixel in space, and effectively reduce the time cost of parameter optimization. Compared with the default parameter scheme, the pixel-level optimization has significantly improved the LAI and GPP estimates. In the next experiment, we will try to use multiple sets of remote sensing products (e.g., BESS, VPM, FLUXCOM, MODIS for GPP), and use the triple-collocation method to give the spatial distribution of the error variance of the products, and take this error into account in the parameter optimization.

We add these analyses in our manuscript, please check lines 509-521. The figure above can be found in our supplement document (Figure S6).

BESS: GPP derived from Breathing Earth System Simulator (BESS) (Jiang and Ryu, 2016);
VPM: Vegetation Photosynthesis Model (Zhang et al., 2017);
FLUXCOM: GPP derived from machine learning methods (Jung et al., 2011);
MODIS : GPP derived from the MOD17A2H V6 product (Running et al., 2004)

References:

[1]. Jiang, C. and Ryu, Y.: Multi-scale evaluation of global gross primary productivity and evapotranspiration products derived from Breathing Earth System Simulator (BESS), Remote Sensing of Environment, 186, 528-547, 10.1016/j.rse.2016.08.030, 2016.

[2]. Jung, M., Reichstein, M., Margolis, H. A., Cescatti, A., Richardson, A. D., Arain, M. A., Arneth, A., Bernhofer, C., Bonal, D., Chen, J., Gianelle, D., Gobron, N., Kiely, G., Kutsch, W., Lasslop, G., Law, B. E., Lindroth, A., Merbold, L., Montagnani, L., Moors, E. J., Papale, D., Sottocornola, M., Vaccari, F., and Williams, C.: Global patterns of land-atmosphere fluxes of carbon dioxide, latent heat, and sensible heat derived from eddy covariance, satellite, and meteorological observations, Journal of Geophysical Research, 116, 10.1029/2010jg001566, 2011.

[3]. Ma, H. and Liang, S.: Development of the GLASS 250-m leaf area index product (version 6) from MODIS data using the bidirectional LSTM deep learning model, Remote Sensing of Environment, 273, 10.1016/j.rse.2022.112985, 2022.

[4]. Running, S. W., Nemani, R. R., Heinsch, F. A., Zhao, M. S., Reeves, M., and Hashimoto, H.: A continuous satellite-derived measure of global terrestrial primary production, Bioscience, 54, 547–560, https://doi.org/10.1641/00063568(2004)054[0547:acsmog]2.0.co;2, 2004.

[5]. Zhang, Y., Xiao, X., Wu, X., Zhou, S., Zhang, G., Qin, Y., and Dong, J.: A global moderate resolution dataset of gross primary production of vegetation for 2000-2016, Scientific data, 4, 170165, 10.1038/sdata.2017.165, 2017.

[6]. Zheng, Y., Shen, R., Wang, Y., Li, X., Liu, S., Liang, S., Chen, J. M., Ju, W., Zhang, L., and Yuan, W.: Improved estimate of global gross primary production for reproducing its long-term variation, 1982–2017, Earth System Science Data, 12, 2725-2746, 10.5194/essd-12-2725-2020, 2020.

3.  In Line 228, the ten sensitive parameters derived in the study are sensitive parameters for which indicators (GPP, LAI, or ER, NEE?)?

**Response:**

Since we used LAI and GPP from the GLASS products as our references when optimizing model parameters, we conducted sensitivity analysis with the target variables of LAI and GPP, and selected the parameter with higher sensitivity index to be optimized (Fig. 1), including seven LAI-sensitive parameters (vmax, p5, p14, p15, p17, p18 and p20) and seven GPP-sensitive parameters (alpha3, theta3, beta3, vmax, p5, p14 and p20). Therefore, there are mainly 10 parameters to be optimized. Considering p5, p14 and p20 have more obvious effect on LAI, we only used optimized results of alpha3, theta3, beta3 and vmax as the known conditions when beginning the second step optimization.

We have modified the corresponding part in our manuscript, please check lines 239-240 and lines 249-251.

4. The ranges of parameters are very important because they directly limit the boundaries of parameter optimization and the range of parameter space variation. Therefore, how did the authors determine the ranges of the prior parameters in this study?

**Response:**

We determined the ranges of the prior parameters according to some published studies. For parameters in the IBIS model, you can find the possible ranges from Cunha et al. (2013) and Varejão et al. (2013). For parameters in the DALEC model, you can follow the descriptions from Bloom et al. (2016), Chuter et al. (2015) and Lu et al. (2017).

We added these references in our manuscript, please check lines 233-234.

References:

[1]. Bloom A. Anthony, J.-F. E., Ivar R. van der Velded: The decadal state of the terrestrial carbon cycle: Global retrievals of terrestrial carbon allocation, pools, and residence times, PNAS, 10.1073/pnas.1515160113, 2016.

[2]. Chuter A.M., P. J. A., Anne C. Skeldon, and Ian Roulstone: A dynamical systems analysis of the data assimilation linked ecosystem carbon (DALEC) models, CHAOS, 2015.

[3]. Cunha, A. P. M. A., Alvalá, R. C. S., Sampaio, G., Shimizu, M. H., and Costa, M. H.: Calibration and Validation of the Integrated Biosphere Simulator (IBIS) for a Brazilian Semiarid Region, J. Appl. Meteorol. Climatol., 52, 2753-2770, 10.1175/jamc-d-12-0190.1, 2013.

[4]. Lu, D., Ricciuto, D., Walker, A., Safta, C., and Munger, W.: Bayesian calibration of terrestrial ecosystem models: a study of advanced Markov chain Monte Carlo methods, Biogeosciences, 14, 4295-4314, 10.5194/bg-14-4295-2017, 2017.

[5]. Varejão, C. G., Costa, M. H., and Camargos, C. C. S.: A multi-objective hierarchical calibration procedure for land surface/ecosystem models, Inverse Probl. Sci. Eng., 21, 357-386, 10.1080/17415977.2011.639453, 2013.

5. Lines 337-338, as you mentioned, for the testing set, the estimated errors (RMSE and DISO) using XGBoost were slightly less correlated with the corresponding accuracy indexes of the MASM approach. However, the input of the target values in Xboost's training set are the parameter values obtained by MASM. So is the training effect of the Xboost model unsatisfactory?

**Response:**

Although the accuracy of the validation data is slightly reduced compared to the training data, the overall difference is not large. On the one hand, it may be because the optimal parameters for training XGBoost may not be the best choice as we took the mean of the posterior distribution as our optimal choice, which could introduce uncertainties, but we have ensured the reliability of the dataset used for training XGBoost through the screening of training data as much as possible (Please see Section 4.2.2). On the other hand, the sensitivity of parameters to environmental variables and target variables is also likely to have spatial differences, which will lead to the inapplicability of parameter estimation results in some regions with low sensitivity (Please see Section 5.2). In addition, the number of samples is also an important factor affecting the training accuracy, and we will consider this issue in our further research.

6. Both MASM and Xboost can be used to obtain spatial distribution of sensitivity parameters. Introducing XGBoost to predict other spatial parameters from some partially corrected parameters may cause greater uncertainty. So why did the authors choose these two methods? It seems that the necessity of two-step correction is not clear in the article.

**Response:**

Although the MASM method can greatly speed up the process of parameter optimization by means of a surrogate model and reduce the calculation time, we still need hundreds of parameter samples and simulations of the original model to provide enough training data for building the surrogate model (GPR model). It is obviously not realistic to implement the MASM method pixel by pixel at higher spatial resolutions, as it still consumes a lot of computation. Therefore, we consider to first select a certain number of samples to implement the MASM algorithm, and then use the XGBoost model to expand the spatial scale. In this way, the results of our first step can provide more samples for training the XGBoost model than directly using machine learning for spatial expansion. Compared to experiments that simply use surrogate models to optimize parameters, our second-step machine learning extension makes it easier to obtain large-scale, high-resolution parameter space distributions.

We have added a corresponding explanation to this question in our manuscript, please check lines 102-105.

7. Parameter screening and optimization mainly targeted at LAI and GPP, while ER and NEE were added to the final carbon flux prediction. NEE means GPP minus ecosystem respiration and disturbance. I wonder if the parameters associated with these two processes were corrected? In addition, suggest to analyze the uncertainties of these two carbon flux results.

**Response:**

The main purpose of our analysis of ER and NEE was to see if optimization of GPP and LAI would have an impact on ER and NEE. On the one hand, the increased accuracy of GPP will improve the simulation of NEE to a certain extent, on the other hand, although we only perform sensitivity analysis for LAI and GPP, and optimize some key parameters. However, parameters that are affected in LAI and GPP optimization, such as turnover, can also affect the simulation of respiration.

In fact, the accuracy of ER and NEE was not significantly improved based on the site-level validation (Fig. S2; Fig 10). The reason for this may be that, the size of respiration is related to carbon pool size, while factors related to forest age and carbon pools are not considered in the current experiment.

In the next step, we will try to integrate forest age and biomass products to improve the key parameters of terrestrial carbon pools (such as allocation, turnover rate, and respiration rate), so that we can improve the simulation accuracy of respiration, and then obtain a more accurate distribution of vegetation carbon storage.

We have added this discussion in our manuscript, please check lines 502-507.

The uncertainty analysis of the model output is an important part of model evaluation, but since we only take the mean of the posterior distribution of the parameters as the optimal value to input into the model to obtain the optimized LAI and GPP, there is no quantitative analysis of model uncertainty. We have done a lot of discussions on the sources of uncertainty in the discussion section, such as the parameter selection, algorithmic and references (remote sensing products) uncertainties. We will add uncertainty quantification as an important part to our experiments in further research.